



# The SahulCHAR Collection: A Palaeofire Database for Australia, New Guinea, and New Zealand

Emma Rehn[1], Haidee Cadd[2], Scott Mooney[3], Tim J. Cohen[2], Henry Munack[2], Alexandru T. Codilean[2], Matthew Adeleye[4], Kristen K. Beck[5], Mark Constantine IV[2], Chris Gouramanis[6], Johanna M. Hanson[7], Penelope J. Jones[8], A. Peter Kershaw[9], Lydia Mackenzie[10], Maame Maisie[3], Michela Mariani[11], Kia Matley[12], David McWethy[13], Keely Mills[14], Patrick Moss[15], Nicholas R. Patton[16,17], Cassandra Rowe[1], Janelle Stevenson[18], John Tibby[19], Janet Wilmshurst[20]

[1]ARC Centre of Excellence for Australian Biodiversity and Heritage, College of Arts, Society, and Education, James Cook University, Cairns, 4870, Australia
[2]ARC Centre of Excellence for Australian Biodiversity and Heritage, School of Earth, Atmospheric, and Life Sciences, University of Wollongong, Wollongong, 2500, Australia
[3]ARC Centre of Excellence for Australian Biodiversity and Heritage, School of BEES, University of New South Wales, Sydney, 2052, Australia
[4] Department of Geography, University of Cambridge, Cambridge CB2 1DB, Cambridgeshire, United Kingdom
[5]Catchments and Coasts Research Group, Department of Geography, University of Lincoln, Brayford Pool, Lincoln, LN6 7TS, United Kingdom
[6]Research School of Earth Sciences, The Australian National University, Canberra, Australia, 0200, Australia
[7]School of Earth and Environment, University of Canterbury, Christchurch, 8041, New Zealand
[8]Menzies Institute for Medical Research, University of Tasmania, Hobart, 7000, Australia
[9]School of Earth, Atmosphere and Environment, Monash University, Clayton, 3800, Australia
[10]School of Geography, Planning and Spatial Sciences, University of Tasmania, Hobart, 7001, Australia
[11]School of Geography, University of Nottingham, NG72RD, United Kingdom
[12]School of BioSciences, The University of Melbourne, Parkville, 3010, Australia
[13]Department of Earth Sciences, Montana State University, Bozeman, Montana 59715, USA
[14]British Geological Survey, Keyworth, Nottingham NG12 5GG, United Kingdom
[15]School of Earth & Atmospheric Sciences, Queensland University of Technology, Brisbane, 4072, Queensland, Australia
[16]Department of Geosciences, Idaho State University, Pocatello, ID, USA
[17]School of Earth and Environment, University of Canterbury, Christchurch, 8041, New Zealand
[18]ARC Centre of Excellence for Australian Biodiversity and Heritage, School of Culture, History and Language, The Australian National University, Canberra, 2601, Australia
[19]Geography, Environment and Population, University of Adelaide, Adelaide, 5005, Australia
[20]Manaaki Whenua - Landcare Research, PO Box 69040, Lincoln, 7640, New Zealand

*Correspondence to*: Haidee Cadd (haidee@uow.edu.au)

**Non-technical summary (max. 500 characters):** This paper presents SahulCHAR, a new collection of palaeofire (ancient fire) records from Australia, New Guinea, and New Zealand. SahulCHAR Version 1 contains 687 records of sedimentary charcoal or black carbon, including digitized data, records from existing databases, and original author-submitted data. SahulCHAR is a much-needed update on past charcoal compilations that will also provide greater representation of records from this region in future global syntheses to understand past fire.





**Abstract.** Recent global fire activity has highlighted the importance of understanding fire dynamics across time and space,
with records of past fire (palaeofire) providing valuable insights to inform current and future management challenges. New
records from the recent increase in palaeofire studies from Australia and surrounds have not been captured in any database
for broader comparisons, and Australasia is poorly represented in current international databases used for global modelling
of palaeofire trends. These problems are addressed by SahulCHAR, a new collection of sedimentary charcoal and black
carbon records from Sahul (Australia, New Guinea, and offshore islands) and New Zealand. Data are stored in the
OCTOPUS relational database platform, with a structure designed for compatibility with the existing Global Paleofire
Database. Metadata are captured at site-level and observation-level, with observations including age determinations and
charcoal or black carbon data. SahulCHAR Version 1 contains 687 records of charcoal or black carbon, including digitized
data, unchanged and modified records from the Global Paleofire Database, and original author-submitted data. SahulCHAR
is a much-needed update on past regional palaeofire compilations that will also provide greater representation of records
from Sahul and New Zealand in future global syntheses.
**Graphical abstract:**

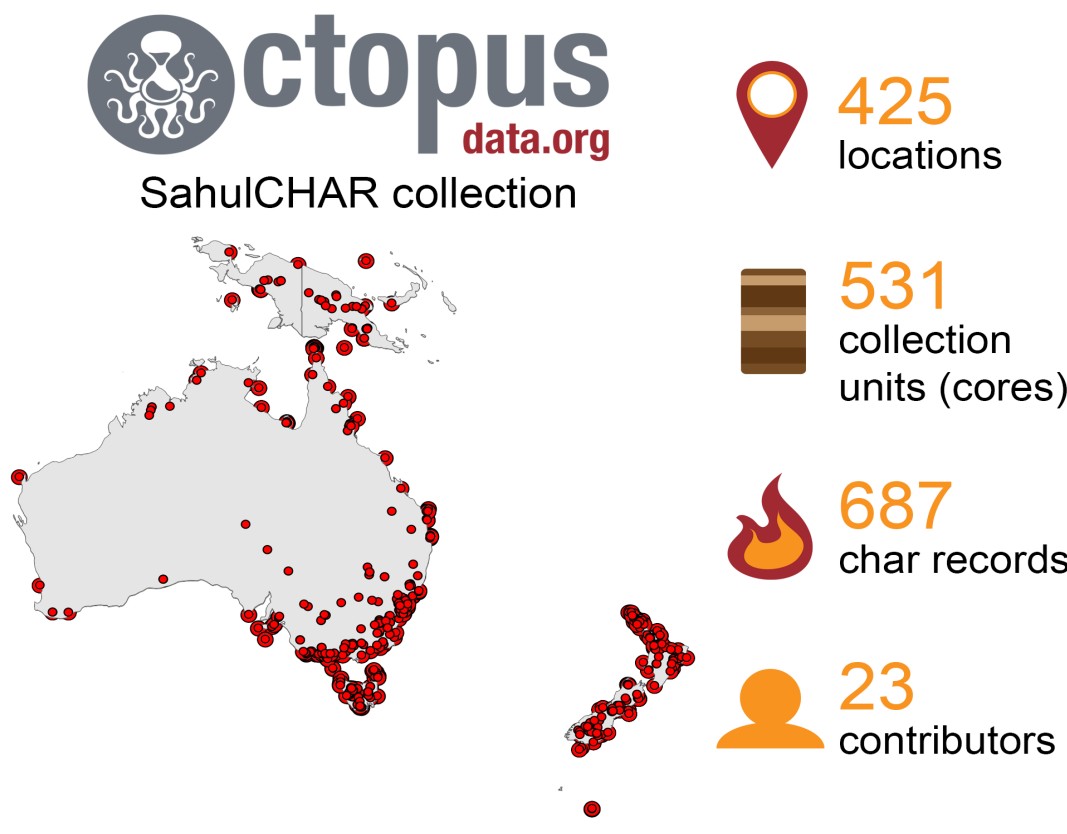




## 1 Introduction

Fire is a key ecosystem process with characteristics that vary widely across biomes globally, with "fire-dependent" ecosystems covering around half of the terrestrial globe (Shlisky et al., 2007, p. 6). Recent increases in global fire activity, including extreme fires in Australia and elsewhere, have highlighted the critical importance of understanding fire dynamics across time and space (Duane et al., 2021; Nolan et al., 2021), with more extreme fire weather predicted in the future for south-eastern Australia and northern and eastern New Zealand as a result of climate change (Lawrence et al., 2022).

Palaeofire data (sedimentary charcoal and black carbon) can offer important insights into past fire variability to inform current and future challenges, including climate-fire-vegetation interactions (Williams & Abatzoglou, 2016; Marlon, 2020). Compilations of palaeofire records have been used to investigate long term relationships and shifting dynamics between humans, fire, vegetation, and climate in Australia (Lynch et al., 2007; Enright & Thomas, 2008; Williams et al., 2015), New Zealand (McWethy et al., 2010; Perry et al., 2014), and Indonesia and Papua New Guinea (Haberle et al., 2001). Understanding fire regimes over long timescales in Australia and the surrounding region has increasingly become a research priority, reflected in a recent influx of new palaeofire records (for examples from just the previous two years, see Adeleye et al., 2023; Constantine et al., 2023; Hanson et al., 2022; Laming et al., 2022; Patton et al., 2023; Rowe et al., 2022; Thomas et al., 2022). However, the last major compilation and synthesis of sedimentary charcoal records from Australasia was Mooney et al. (2011, 2012), containing 224 sedimentary charcoal records, primarily derived from the Global Paleofire Database (GPD, formerly known as the Global Charcoal Database; Power et al., 2010). More recent syntheses have been focused on specific regions, such as Mariani et al.'s (2022) compilation of over 100 charcoal records from south-eastern Australia contained in the GPD to investigate human-fire-vegetation dynamics over the last thousand years. The diverse environments of Australia, New Guinea, and New Zealand have unique histories of fire-climate-human interactions (Mooney et al., 2011). As identified by Mooney et al. (2011) and Rowe et al. (2023), no individual palaeofire record should be considered representative of this vast region; to disentangle long-term influences on fire and potential variations across subregions and ecosystems, a large dataset is required.

Major global databases containing charcoal data such as the GPD, the Reading Palaeofire Database (RPD; Harrison et al., 2022), Neotoma Paleoecology Database (Neotoma; Williams et al., 2018), and PANGAEA (Feldner et al., 2023) are lacking many palaeofire records from Australasia. The GPD currently contains 179 cores with associated charcoal data from Australia, 23 cores from New Guinea (Papua New Guinea and West Papua), and 10 cores from New Zealand. These records are replicated in the RPD, and even fewer cores from this region are available in Neotoma (17 cores) and PANGAEA (11 cores, 6 of which are also contained in the GPD). The GPD is a valuable resource for regional and global palaeofire syntheses (e.g., Daniau et al., 2012; Karp et al., 2021; Marlon et al., 2013, 2016) but it requires a significant update to capture the many new palaeofire records now available from Australia, New Guinea, and New Zealand. As noted by Harrison et al. (2022), current limitations of the GPD include potential duplicates of sites, missing metadata and age data,



and necessary updates to incorporate newly published records. The GPD also contains lengthy yet still incomprehensive lists
of metadata options, in part due to the array of ways to approach charcoal analysis (e.g., Mooney & Tinner, 2010; Turner et
al., 2004) as well as ad hoc user additions and structural constraints (most notably a single field for measurement units that
includes size ranges).
These limitations are addressed for Sahul (Australia, New Guinea, and continental islands) and New Zealand by the
SahulCHAR data collection (Rehn et al. 2024). The purpose of this paper is to introduce the data structure of SahulCHAR
and provide an overview of data compiled in Version 1. In keeping with existing OCTOPUS collections, SahulCHAR was
named and intended to have a geographic focus on the Sahul landmass (Australia, New Guinea, and continental islands);
during data collection, the geographic scope was extended to include New Zealand. SahulCHAR data collection was
designed to capture new records published since the compilation by Mooney et al. (2011, 2012), to capture older records not
previously entered in the GPD, to check (and correct, if required) details of records in the GPD from this region, and to
capture additional metadata wherever possible for records available in the GPD.

## 100  2 Data structure and compilation

SahulCHAR is hosted on the OCTOPUS platform (https://octopusdata.org), with data stored in a relational PostgreSQL
database (Figure 1). For a full description of the OCTOPUS v.2 system architecture, see Codilean et al. (2022), and for a full
and up-to-date OCTOPUS documentation, see Munack et al. (2023). For compatibility and future integration, the data
structure of SahulCHAR is broadly based on the metadata captured in the Global Paleofire Database (GPD) and structured to
comply with requirements of the OCTOPUS platform. Data in SahulCHAR are captured at SITE, UNIT, SAMPLE and
OBSERVATION levels, with not all UNIT types being cores (e.g., archaeological excavations, sediment monoliths).
Following the protocol used by the GPD, unnamed cores are assigned a CORE name consisting of their associated site name
and the suffix '_core1', '_core2', etc.







**Figure 1:** Graphical representation of the OCTOPUS semantic database model featuring the fully integrated SahulCHAR partner collection. SahulCHAR shares parent/lookup tables with the other collections (SahulArch, SahulSED, the IPPD, FosSahul, ExpAge, and CRN) on global, regional, and bibliographic level.



114  **Table 1:** Site-level metadata collected in SahulCHAR.

| Metadata field | Description | Field type | Example | Corresponding GPD field |
|---|---|---|---|---|
| METASITE | Metasite name | Free text | Big Willum Swamp | *NA* |
| SITE | Site name | Free text | Big Willum Swamp BWIL2 | site_name |
| COUNTRY | Country where metasite is located | Predefined list | Australia | country_name |
| SITECODE | Site type, based on primary characteristics at the time of collection | Predefined list | terrestrial, bog | site_type_desc |
| BASIN | Basin size | Predefined list | large (50.1-500 km²) | basin_size_desc |
| CATCHMENT | Catchment size | Predefined list | small (<10 km²) | catchment_size_name |
| FLOWTYPE | Water flow type | Predefined list | closed - no inflow or outflow | flow_type_name |
| BIOME | Surrounding biome type | Predefined list | For full list, https://octopus-db.github.io/documentation/ | biome_type_name |
| CORE | Name of collection unit, such as a core or excavation square | Free text | BWIL2 | core_name |
| X_WGS84 | Longitude | Numeric (in decimal degrees) | 141.998466 | longitude |
| Y_WGS84 | Latitude | Numeric (in decimal degrees) | -12.656479 | latitude |
| ELEVATION | Elevation above sea level | Numeric (in metres) | 28 | elevation |
| CORDS_ELEV | Source of coordinates and elevation data | Predefined list | INTP_INTP | *NA* |
| WATERDEPTH | Water depth at time of sampling | Numeric (in metres) | 3.5 | water_depth |
| COREDATE | Sampling date | Date (dd/mm/yyyy) | 01/07/2017 | coring_date |
| CORETYPE | Method used to collect the sample | Predefined list | piston corer | core_type |
| DEPOS_TYPE | Depositional context type | Predefined list | alluvial sediment | depo_context |

115



## 2.1 Site-level metadata

Site-level metadata fields, descriptions, and examples are presented in Table 1; for complete documentation including available options for predefined lists, see https://octopus-db.github.io/documentation/data_tables.html#global-georeferencing-tables and https://octopus-db.github.io/documentation/data_tables.html#non-cosmogenics-tables. Location data are captured in two forms: metasites and sites. Metasites are area-based (such as a lake) and stored as polygons, while sites are point-based (such as a specific coring location in a lake) and stored as coordinates in decimal degrees. Metasites may have multiple associated sites.

Basin and catchment metadata in SahulCHAR (BASIN and CATCHMENT) have been limited to broad categories that do not require numeric values as these data are not often known. Vegetation metadata were limited to broad categories for the major biome surrounding the site (BIOME) as multiple vegetation fields would require extensive list options to be comprehensive. The available options for predefined lists were based on options available in the GPD, with additions where necessary; these changes were informed by author-submitted data.

## 2.2 Unit to Observation-level metadata

Fields shared across all observation-level data are CORE (core or sample name), OBSID1 (internal OCTOPUS identifier, incorporating CORE and identified as 'char' or 'age'), SMPID (internal OCTOPUS identifier, incorporating CORE and DEPTH), DEPTH, THICKNESS, and references (REFDBID). Observation-level data include ages and charcoal or black carbon records.

### 2.2.1 Age metadata

Age metadata collected in SahulCHAR are presented in Table 2; for complete documentation including available options for predefined lists, see https://octopus-db.github.io/documentation/data_tables.html#sahulchar-tables. The predefined list options are based on options available in the GPD, with the exception of the METHOD field which uses an existing OCTOPUS parent table (see https://octopus-db.github.io/documentation/parent_tables.html#cabah-methodid-fields) to allow for a larger range of options. In line with existing OCTOPUS collections of radiometric ages (such as SahulArch; Saktura et al., 2023), during data entry for Version 1, preference was given to uncalibrated rather than calibrated radiocarbon ages where possible, to allow for recalibration with future calibration curve updates. Ages reported in calendar years BC/AD or BCE/CE were converted to 'years BP' prior to entry or entered as AGE_UNIT = 'other' if conversion is not possible. Ages generated from dating methods that are measured as years prior to sample collection and do not require calibration, such as lead-210 or optically stimulated luminescence, were converted to 'years BP' prior to entry where possible or entered as AGE_UNIT = 'other'.



**Table 2:** Age metadata collected in SahulCHAR.

| Metadata field | Description | Field type | Example or available list | Corresponding GPD field |
|---|---|---|---|---|
| CORE | Name of collection unit, such as a core or excavation square | Free text | BWIL2 | core_name |
| OBSID1 | Unique identifier for observation | Text | BWIL2_0.05_age | NA |
| SMPID | Unique identifier for sample | Text | BWIL2_0.05 | id_sample |
| DEPTH | Sample depth (mid-point) in metres | Numeric (in metres) | 0.01 | depth_value |
| THICKNESS | Sample thickness in centimetres | Numeric (in centimetres) | 1 | *NA* |
| LABID | Laboratory ID code for age | Free text | OZX-211 | laboratory number |
| AGE | Age value | Numeric | 760 | age_value |
| AGE_ERROR | Age error value | Numeric | 20 | *NA* |
| AGE_UNIT | Measurement unit for age and age error | Predefined list | radiocarbon years BP | age_units_type |
| METHOD | Dating method used to generate age | Predefined list | Radiocarbon dating | age_type_name |
| MATERIAL | Material dated | Predefined list | bulk sediment, peat | Mat_dated_type |
| REFDBID1, REFDBID2, REFDBID3 | A unique identifier for associated references using the surname of the first author, year of publication, and a keyword (Name:YEARkeyword) | Text | Rehn:2020thesis, Rehn:2021cape | *NA* |

### 2.2.1 Charcoal and black carbon metadata

Charcoal and black carbon metadata collected in SahulCHAR are presented in Table 3; for complete documentation including available options for predefined lists, see https://octopus-db.github.io/documentation/data_tables.html#sahulchar-tables. Charcoal and black carbon (hereafter referred to collectively as 'char') observations may share the same SMPID as age observations, if they are taken from the same depth. Predefined lists are based on options available in the GPD, except



for the CALCURVE and CALPROGRAM fields, as the closest corresponding fields in the GPD
('calibration_curve_version' and 'calibration_method_type', respectively) appear as blank dropdowns in the GPD data
upload interface and contain no values in data exports.
The structure of SahulCHAR differs from the GPD in its approach to char sizes and measurement units. Char particle sizes
in the GPD are embedded within the field for measurement units ('charcoal_units_name'), resulting in a lengthy (176
options) but incomplete list of available units. To address this limitation, char sizes in SahulCHAR are distinct from
measurement units (CHARMEASURE) and entered separately as maximum (CHARMAX) and minimum values
(CHARMIN), along with the measurement unit for these size values (CHARSIZE_U). This allows for a restricted (35
options) yet comprehensive list of measurement units that can be paired with any combination of size values, which may
then be merged into a single field during data migration to the GPD. This database structure also allows users to easily
separate records by size values for analysis.
While the CHARMEASURE field allows for a wide range of measurement units, volumetric (e.g., fragments/cm^3) rather
than influx (e.g., fragments/cm^2/year) measurements for char were preferred where possible during data compilation to
allow for recalibration and recalculation of age-depth models when necessary.
SahulCHAR contains char data from the following sources: original data contributed directly by authors; original data,
digitized data, and data of unknown origins contained in the GPD; original data contained in non-GPD databases (Neotoma
and PANGAEA); original data available in published supplementary materials; and records digitized from published
diagrams. Original char data from any source are classified in SahulCHAR as CHARSOURCE = 'author'. Data sourced
from another database either from digitization data or unknown origins are classified as 'paleofire database', and data
digitized from published diagrams for SahulCHAR are classified as 'digitized'. Data were manually digitized for
SahulCHAR using WebPlotDigitizer (Rohatgi, 2022). Char data from the GPD were exported via the web interface on 27
February 2023. Char data were last accessed from PANGAEA on 25 May 2023 and from Neotoma on 11 July 2023. While
the RPD contains Australasian char data, these records are derived from the GPD and therefore the RPD was not used for
SahulCHAR data compilation.
**Table 3:** Charcoal and black carbon metadata collected in SahulCHAR.

| Metadata field | Description | Field type | Example or available list | Corresponding GPD field |
|---|---|---|---|---|
| CORE | Name of collection unit, such as a core or excavation square | Free text | BWIL2 | core_name |
| OBSID1 | Unique identifier for | Text | BWIL2_char1_1 | |





| | observation | | | |
|---|---|---|---|---|
| SMPID | Unique sample identifier | Text | BWIL2_0.05 | id_sample |
| DEPTH | Sample depth (mid-point) in metres | Numeric (in metres) | 0.01 | depth_value |
| THICKNESS | Sample thickness in centimetres | Numeric (in centimetres) | 1 | *NA* |
| EST_AGE | Estimated age for sample | Numeric (in years BP) | 350 | est_age_cal_bp |
| CALCURVE | Calibration curve used to generate estimated sample age | Predefined list | SHCal20 | calibration_curve_version |
| CALPROGRAM | Calibration program used to generate estimated sample age | Predefined list | rbacon 2.3.2 | calibration_method_type |
| CHARCOUNTS | Charcoal or black carbon count | Numeric | 0.52 | quantity |
| CHARMETHOD | Preparation method used for charcoal or black carbon analysis | Predefined list | pollen slide | charcoal_method_name |
| CHARMEASURE | Measurement units for charcoal or black carbon counts | Predefined list | frag/cm^3 | charcoal_units_name |
| CHARMAX | Maximum size for charcoal or black carbon | Numeric | 250 | *NA (included in 'charcoal_units_name')* |
| CHARMIN | Minimum size for charcoal or black carbon | Numeric | 125 | *NA (included in 'charcoal_units_name')* |
| CHARSIZE_U | Size units for maximum and minimum sizes | Predefined list | μm | *NA (included in 'charcoal_units_name')* |
| CHARSOURCE | Source of charcoal or black carbon data (in field CHARCOUNTS) | Predefined list | author | data_source_desc |
| REFDBID1, REFDBID2, REFDBID3 | A unique identifier for associated references using the surname of the first author, year of publication, and a keyword (Name:YEARkeyword) | Text | Rehn:2020thesis, Rehn:2021cape | *NA* |

## 3 Data summary


SahulCHAR Version 1 (V1; Rehn et al. 2024) contains 687 charcoal and black carbon ('char') records from 531
cores/samples (hereafter referred to as 'cores'), derived from 425 metasite locations across Sahul (Australia, New Guinea,
and the Aru Islands) and New Zealand (Figure 2). The majority of metasites are from Australia (~64%), followed by New
Zealand (~29%). Metasites show some geographic clustering, particularly in south-eastern Australia and the New Guinea
Highlands, with large spatial gaps in central, western, and parts of northern Australia.

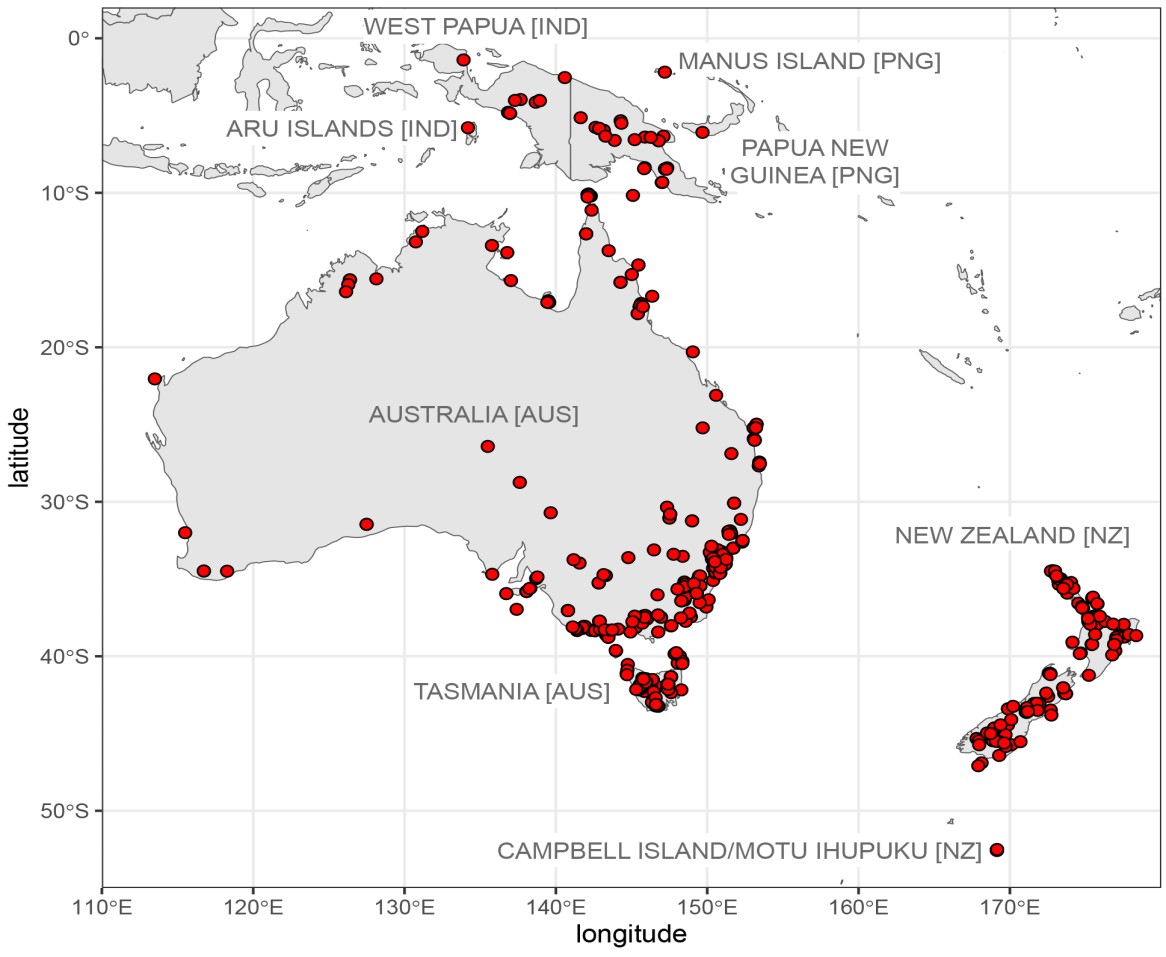


**Figure 2: Sites with charcoal or black carbon records contained in SahulCHAR Version 1, with labels identifying**
**major islands and their national affiliation in square brackets. Nation abbreviations: AUS: Australia, IDN:**
**Indonesia, NZ: New Zealand, PNG: Papua New Guinea.**

Original data were contributed directly to SahulCHAR by 23 authors, totalling 141 records. In cases where author-submitted
data overlap with records that already exist in the GPD (27 records), preference was given to the author-submitted versions.
Approximately 33% (211) of the records in SahulCHAR derive from, or also exist in, the GPD, with 85 records modified in
some way (such as additional or corrected metadata) with reference to author-submitted information or source publications
(Figure 3). Approximately 46% of records in SahulCHAR are digitized from published diagrams.

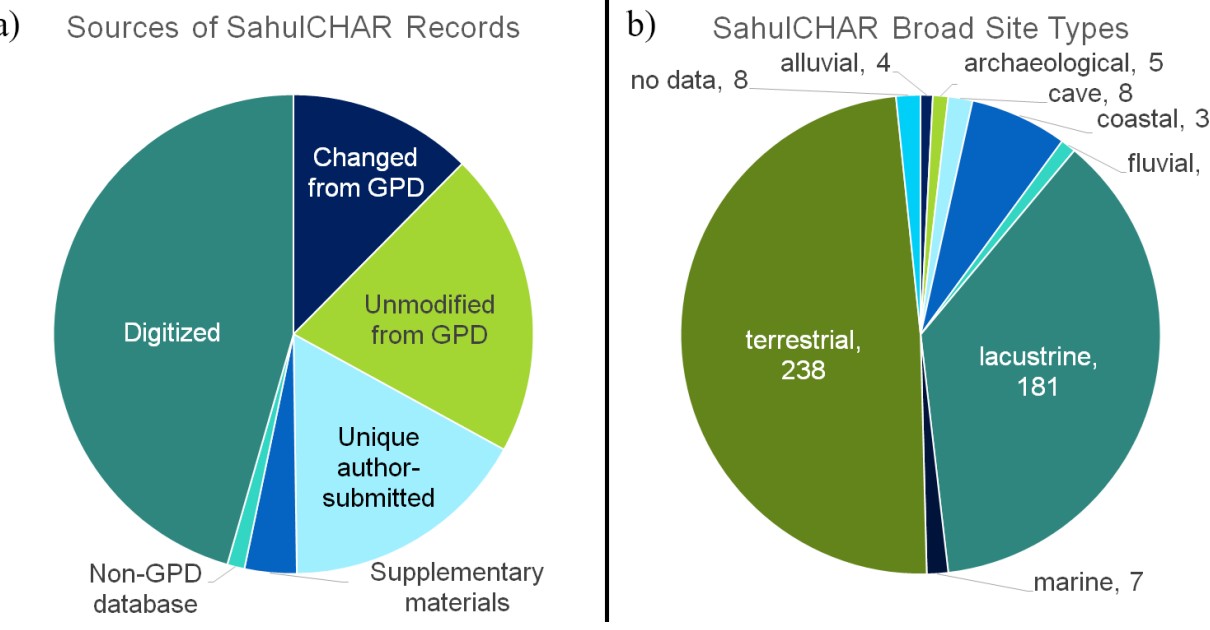


**Figure 3: SahulCHAR V1 a) sources of char records, and b) broad site types.**
The options for SITECODE in SahulCHAR include broad types (e.g., 'terrestrial', 'lacustrine') and broad types with specific
subcategories (e.g., 'terrestrial, bog', 'terrestrial, fen') stored in a self-referencing table with subcategories linked to their
next common denominators, respectively; for ease of comparison, sites are grouped into broad types in Figure 2b. Most sites
in SahulCHAR are broadly categorised as terrestrial (~49%, 238 sites), primarily bogs (107 sites), followed by lacustrine
(~37%, 182 sites), primarily classified as lacustrine with no subcategories (154 sites). Categories in SITECODE are not
exclusive and may overlap (e.g., coastal lakes may be classified as coastal or lacustrine), with these classifications intended
as a general guide. Site characteristics may also change through time; SITECODE was determined based on site
characteristics at the time of sample collection. While archaeological sites were included, these were limited to charcoal
quantification undertaken as part of palaeoenvironmental analyses to exclude charcoal potentially associated with
archaeological features (e.g., hearths). These archaeological sites were further limited to records where associated depth
values were available for char measurements; archaeological sites with char data associated with stratigraphic (SU) or
excavation (XU) units without specified depths were excluded.

Earth System  Discussions
Open Access  Science
Data



**Figure 4: SahulCHAR V1 char data summaries: a) number of char records per UNIT with instances of multiple records from a single core representing different char sizes, analysis methods, or measurement units, b) sample preparation method for char records, c) measurement units (for a full list of measurement units and associated abbreviations, see https://octopus-db.github.io/documentation/parent_tables.html#global-varunitid-fields), and d) number of entries (sample depths) per char record.**



A total of 3271 ages are contained in SahulCHAR V1. The majority (~77%) of cores have 1-10 associated ages, and 34 units (~6%) have no available age data. In instances where no ages are available from a unit with associated charcoal data, other dated units from the same site have been included where possible (5 units, from metasites Lake George and Blue Lake Kosciuszko, both in New South Wales, Australia).

Most UNIT entries have one associated char record (418 cores, ~79%) up to a maximum of six associated records (1 core, 'MAR2' from metasite Marura) (Figure 4). The majority (~63%, 432 records) of char records in SahulCHAR are derived from pollen slides, followed by sieved samples (~32%, 217 records). Pollen slide charcoal also dominated the dataset compiled by Mooney et al. (2012), although sieved charcoal is slightly better represented in SahulCHAR (compared to ~80% and ~20%, respectively, in Mooney et al., 2012, p. 18). Approximately 32% (220 records) of the char records in SahulCHAR are measured in "frag/cm^3", followed by "% of pollen sum" (~14%, 95 records) and "frag/cm^2/yr" (~14%, 94 records). Over half (~54%) of the char records specify a size range for particles, with 54 unique size ranges specified; this demonstrates both the utility of isolating maximum and minimum particle sizes from measurement units to allow for this variability, and the diversity of approaches used to create these records. All char records contain a minimum of three entries, and most char records (~67%) contain 50 entries or less. The highest number of entries for any char record is 881 ('WL15-2_char1' from Welsby Lagoon).

## 4 Conclusions and future work

SahulCHAR is the most comprehensive and up-to-date palaeofire database for Sahul and New Zealand (Rehn et al. 2024), and an overdue step towards improved representation of Australasia in global syntheses. The latter goal will be addressed through upcoming integration with the GPD as part of the planned conversion of the GPD into a constituent database of Neotoma Paleoecology Database (Dietze and Vannière, 2022). As an update to the last Australasian compilation (Mooney et al., 2011, 2012, which covered a slightly larger geographic area than SahulCHAR), SahulCHAR triples the number of char records available for the region and incorporates data from numerous new studies produced over the last decade. SahulCHAR follows the FAIR principles of scientific data management and stewardship (Wilkinson et al., 2016) and the OPEN data requirements of funding agencies, such as the Australian Research Council, to make publicly funded data freely available.

Data creators in the region are encouraged to contribute records either directly to SahulCHAR or to the GPD within Neotoma. Future versions will ideally shift the balance of char sources away from digitized data, with a greater representation of author-contributed original data. Future work relating to SahulCHAR Version 1 will provide a synthesis and analysis of the records in the dataset to explore trends in palaeofire regimes across the region, and could also explore metadata associated with each record to understand changing approaches to charcoal analysis over time.



Data creators with char records from Australia, New Guinea, or New Zealand that they would like to contribute can use a
SahulCHAR data template (10.5281/zenodo.10117180; Rehn, 2023) and can contact Dr Haidee Cadd (haidee@uow.edu.au)
with enquiries or to submit completed data templates.

**Data availability statement**

The data in this study are openly available at http://dx.doi.org/10.25900/KKDX-XH23 (Rehn et al. 2024) and via the
Octopus database https://octopusdata.org/ (last accessed 28th August 2024; Codilean et al. 2022). Additional information
about the SahulCHAR database collection and the data can be accessed at: http://dx.doi.org/10.25900/KKDX-XH23.

**Funding**

This work was supported by the ARC Centre of Excellence for Australian Biodiversity and Heritage [CE170100015] and
ARC Discovery projects DP150103875 and DP190102782.

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
