# Peer review of "The SahulCHAR Collection: A Palaeofire Database for Australia,"

_Earth System Science Data, 2024_

## Author Response (AR1)

**General comments:**

This paper describes a late-Quaternary sedimentary charcoal database from the Sahul region (Australia, New Guinea, and New Zealand). The data set will be a welcome update for a region where the variations of fire are particularly important for our overall understanding of the controls and history of biomass burning. A considerable amount of effort has been expended in harmonizing metadata and retrieving data from original sources. For the most part, the manuscript is well written and clear. However, there are a few misapprehensions or omissions in the text, and the relationship between the specific data in this database and "legacy" databases (e.g. GPD/paleofire.org) needs to be reviewed and perhaps reconsidered.

My first specific comment below illustrates some of the general issues in the initial development and propagation of databases: misapprehension of the heritage and contents of existing databases (and uncritical acceptance of those contents). This is not a simple matter of attribution, but these issues can have impacts on the scientific results they are used to generate if they are not appreciated by uncritical users. I think the current manuscript and database do a good job of avoiding those issues, but they are not completely free from them.

Consequently, I think a little light editing and an "internal review" of the GPD-to-SahulCHAR data transfer would suffice to make the manuscript publishable, and it would certainly be timely, given the apparent community-wide desire to ingest charcoal data into Neotoma.

**Specific comments:**

p. 3, line 70: "However, the last major compilation and synthesis of sedimentary charcoal records from Australasia was Mooney et al. (2011, 2012), containing 224 sedimentary charcoal records, primarily derived from the Global Paleofire Database (GPD, formerly known as the Global Charcoal Database; Power et al., 2010)."

There are several issues here:  Mooney et al. (2011) used data from the Global Charcoal Database version 2 (GPDv2, Daniau et al. 2012):  ("We extracted 196 sedimentary charcoal records this broadly defined Australasian region (20N - 50S and 100E to 177W) from Version 2 of the Global Charcoal Database (GCD-V2; Daniau et al., in preparation,) compiled by the Global Palaeofire Working Group (GPWG: http://gpwg.org/). These data were supplemented by an additional 27 sites, chosen to increase the number of long records and to improve the spatial coverage (GCD-V2.5).") Mooney et al. (2012) used the same data set ("GCDv2.5"), which was a preliminary version of GCDv3 (Marlon et al., 2013, 2016).

Neither paper therefore used data from the Global Paleofire Database (as represented by the R package GCD (e.g. GCDv4.0.7), or the web interface at paleofire.org, neither of which were in existence in 2011 or 2012). There is a distinction between the various GCD's of Power, Daniau, and Marlon, and the derivative GPD ("GPDv4" available as the R package GCDv4.0.7 or via the web interface at paleofire.org). Although the new database (SahulCHAR) does not directly depend on any of the GCD databases, it does so indirectly through the GPD, which in some instances degraded the data in the antecedent GPDv3 (see comment for p. 3, line 86 below). This kind of fuzziness contributes to mispprehension of the actual sources of the "legacy" data in subsequently produced databases, and the extent to which those data may or may not have problems that may be inherited by subsequent databases. This is a community-wide issue, and not specific to SahulCHAR, but it would be good to straighten things out here before publication.

p. 3, line 82: "These records are replicated in the RPD [RPDv1b] …" "Replication" here might imply "copy", which was not quite the case (Harrison et al., 2022). Many records were given new chronologies, and in some cases the GPD data were replaced by data obtained from original authors.

p. 3, line 86: "As noted by Harrison et al. (2022), current limitations of the GPD include potential duplicates of sites, missing metadata and age data, and necessary updates to incorporate newly published records." In addition, the GPD lost precision in locational metadata and in key variables such as sample depth and charcoal values relative to the parent GCDv3 data. For example, Galway Tarn's location in the GPD (e.g., in GCDv4.0.7), which is inherited here, is given to two decimal places, while in GCDv3 it was given to six. This seems like a negligible difference, but resolves to a distance of about kilometer, which could become significant in matching different data sets or extracting modern climate data for sites.

In the case of sample depth, loss of precision would influence the calculation of influx from concentrations (or the reverse), and this would be compounded if the moribund R paleofire package was used, which defaults to a "spreadsheet" approach for calculating sample thickness. (Explicit sample thickness values that were present in GCDv3 were dropped in GCDv4.) Consequently, it would be important to not uncritically accept data from the GPD, but instead to review what has been copied.

Along those lines, it would be optimal to add an additional field to the metadata, perhaps called "HOWCOPIED" that might have the levels "verbatim" (to indicate data copied without modification), "repaired" (for instances where, for example, depth was recorded in centimeters, when it was actually in meters), "replaced" (for instances when data in the source database that might have been degraded in precision is replaced by author-provided data), and so on. (I don't know exactly what was done in migrating data from the GPD, so there may be more levels/categories.)

Tables 1-3: It would be good at some point to point out that the tables are for discussion here, and are not constituent tables in the database.

p. 9, line 161: "… during data migration to the GPD." I would have reservations about doing that. I think that the SahulCHAR database has several advantages over the current GPD, and should stand alone, or become a constituent of a newer database.

p.9, line 173: "Char data were last accessed from PANGAEA…" The R code unique(sahulchar_sf$DATASOURCE) applied to the sahulchar sf/data.frame returns " "Author", NA, "Digitised", "Paleofire database" ". Are the PANGAEA data folded into one of those categories?

p. 9, line 173: "While the RPD contains Australasian char data, these records are derived from the GPD and therefore the RPD was not used for 175 SahulCHAR data compilation." But RPDv1b does contain alternative chronologies relative to those originally published (as well as other fixups), and so the RPDv1b data are not a simple mirror of the GPD data.

p. 12, lines 188-192: This discussion explains why migrating SahulCHAR to GPD is a bad idea.

Table 3: "CHARCOUNTS" Inspection of the data (and the example in the table) shows that the values in this field are not always counts (which should be integer values), but include influx and concentration values, ratios, etc. To avoid future misapprehensions of what the database contains, I suggest renaming this to "CHARVALUES" or something (but not just "CHAR" which could be ambiguously interpreted as "charcoal accumulation rate").

Figs. 3 and 4: Pie charts are denigrated for the kind of application here.

Fig. 4d: Why the strange class intervals?

**Technical comments:**

I made several attempts to access the data:

*1) Via the web interface at https://octopusdata.org/:*

I was able to open the interface, but when attempting to export data, got the following message, which implies the available data online are only a subset of the database intended to illustrate the kind of metadata available:

"Sedimentary Charcoal & IPPD: The data provided here is a subset, including only overview information at the collection-unit and dataset levels. For complete data access, refer to the online documentation and use WFS (Web Feature Service)."

*2) Via WFS and QGIS, using the link: http://geoserver.octopusdata.org/geoserver/wfs*

Using the QGIS instructions at https://octopus-db.github.io/documentation/usage.html, I was able to open the "Sedimentary Charcoal Records: Sahul & NZ (all data)" collection, but could not find data such as the sample depths and charcoal values, only metadata. This is consistent with the above message.

*3) Using the R instructions at https://octopus-db.github.io/documentation/usage.html*

I was not able to open the collection, getting the error message:

"Cannot open "http://geoserver.octopusdata.org/geoserver/wfs?service=wfs&version=2.0.0&request=GetFeature &typename=Sedimentary%20Charcoal%20Records%3A%20Sahul%20%26%20NZ%20%28all%20data %29&srsName=EPSG%3A900913"; The file doesn't seem to exist."

(I'm guessing that upon acceptance of the paper, these three approaches will "go live". If not, some fixups or better instructions would be in order.)

*4) Via the downloadable shapefile at http://dx.doi.org/10.25900/KKDX-XH23 and R:*

I was able to able to open the file, and export the data as a .csv file, using the following code:

```
library(sf)

shp_file <- "/Users/bartlein/Dropbox/Docs/Rehn/sahulchar/sahulchar.shp"

sahulchar_sf <- st_read(shp_file)

csvfile <- "/Users/bartlein/Dropbox/Docs/Rehn/sahulchar.csv"

write.csv(sahulchar_sf, csvfile)
```

This yielded a .csv file that (it's fair to say) is a little messy. However, simple sorting on site-name and sample-depth fields (or parsing the sf/data.frame using R) would yield data that could be easily incorporated into an analysis.

Thank you for your positive and constructive comments on this manuscript. We hope that this update will be a useful contribution to the Australian and global palaeofire communities.

Response to general comments –

We acknowledge the ambiguity of the text in regards to propagation of previous Palaeofire databases. This was an omission of the correct references here on our part, which we have rectified in the text (detailed below). We have also tried to improve the clarity of the text surrounding the discussion of previous databases and how they were incorporated into the new SahulCHAR database.

We have included additional information in the Supplement that specifies the way each individual site was incorporated into SahulCHAR from the GPD. Very few sites were transferred verbatim and each record was critically appraised, with additional information or corrections added when necessary.

Specific comments -

**p. 3, line 70: "However, the last major compilation and synthesis of sedimentary charcoal records from Australasia was Mooney et al. (2011, 2012), containing 224 sedimentary charcoal records, primarily derived from the Global Paleofire Database (GPD, formerly known as the Global Charcoal Database; Power et al., 2010)."**

We acknowledge we have used the incorrect citations here and have modified the text to make it clear that the data used by Mooney was subsequently included into GPD but this was not where it originated.

"However, the last major compilation and synthesis of sedimentary charcoal records from Australasia was Mooney et al. (2011, 2012), containing 224 sedimentary charcoal records. These records were primarily derived from the Global Charcoal Database version 2 (GPDv2, Daniau et al. 2012) and version 2.5 (GPDv2.5. Marlon et al., 2013, 2016). These records are now included in the Global Paleofire Database (GPDv4, Power et al., 2010)"

**p. 3, line 82: "These records are replicated in the RPD [RPDv1b] …" "Replication" here might imply "copy", which was not quite the case (Harrison et al., 2022). Many records were given new chronologies, and in some cases the GPD data were replaced by data obtained from original authors.**

We have altered this text to read "These records are also contained in the RPD…." To reduce the suggestion that these records are identical in the RPD.

**p. 3, line 86: "As noted by Harrison et al. (2022), current limitations of the GPD include potential duplicates of sites, missing metadata and age data, and necessary updates to incorporate newly published records." In addition, the GPD lost precision in locational metadata and in key variables such as sample depth and charcoal values relative to the parent GCDv3 data.**

We recognise that some data contained in the GPD is not without issues and we believe this comment is largely due to a lack of clarity in the text that suggested we accepted data from GPD uncritically. We have since included additional information to this manuscript and supplement that specifies which sites were copied verbatim (after assessment) from GPD and which sites included additional information either from original or subsequent publications and directly from authors.

"To avoid replicating potential errors contained in the GPD, all data from the GPD were carefully screened during data entry for SahulCHAR. Published sources – including but not limited to sources listed in GPD records – were used to modify and add to metadata and charcoal data captured in the GPD wherever possible, as part of a greater literature review to identify charcoal records in the region."

**Tables 1-3: It would be good at some point to point out that the tables are for discussion here, and are not constituent tables in the database.**

We have changed captions for these tables to indicate that these table are the information that is recorded and available to view/sort for each site in the database. "Table representation of SahulCHAR attributes vs corresponding GDP fields, where applicable. For full description of database tables refer to Munack et al. 2023"

**p.9, line 173: "Char data were last accessed from PANGAEA…" The R code unique(sahulchar_sf$DATASOURCE) applied to the sahulchar sf/data.frame returns " "Author", NA, "Digitised", "Paleofire database" ". Are the PANGAEA data folded into one of those categories?**

All Neotoma and PANGAEA data were entered with the source as 'author' because those databases explicitly show that those records were submitted directly by the authors. Please see line 175 for the intext explanation -

"Original char data from any source are classified in SahulCHAR as CHARSOURCE = 'author'. Data sourced from another database either from digitization data or unknown origins are classified as 'paleofire database', and data digitized from published diagrams for SahulCHAR are classified as 'digitized'"

**p. 9, line 173: "While the RPD contains Australasian char data, these records are derived from the GPD and therefore the RPD was not used for 175 SahulCHAR data compilation." But RPDv1b does contain alternative chronologies relative to those originally published (as well as other fixups), and so the RPDv1b data are not a simple mirror of the GPD data.**

We have made it more clear that the records within these two databases are not simply replica's and have also specified why we have not included updated chronologies for these records.

"While the RPD contains Australasian char data, these records were derived from the GPD. The methodology applied here involved assessing individual records from GPD and modifying, updating or correcting records where necessary based on local knowledge or discussion with original authors. Therefore, even though the RPD includes alternate chronologies and other modifications from GPD, this was not used for SahulCHAR data compilation. Due to constantly evolving and updated chronological modelling and calibration techniques we have not included new chronologies for individual records and recommend re-calibration of age-depth models using the most appropriate up to date methods for records included in SahulCHAR at time of use."

**Table 3: "CHARCOUNTS" Inspection of the data (and the example in the table) shows that the values in this field are not always counts (which should be integer values), but include influx and concentration values, ratios, etc. To avoid future misapprehensions of what the database contains, I suggest renaming this to "CHARVALUES" or something (but not just "CHAR" which could be ambiguously interpreted as "charcoal accumulation rate").**

This is a good point. We will change the name of the CHARCOUNTS to CHARVALUES to include the different charcoal measures included.

**Figs. 3 and 4: Pie charts are denigrated for the kind of application here.**

We have not changed the pie charts here as they are reflecting a subset of the data, so we believe the proportional data representation is ok in this instance.

**Fig. 4d: Why the strange class intervals?**

We have adjusted the class intervals.

**I was not able to open the collection, getting the error message**

This was likely an issue with the code due to unclear instructions in the documentation. It was not made clear that the required typename= "be10-denude:sahulchar" to get the charcoal data.

Unfortunately the 'messiness' of the data is difficult to help with the shear amount of data and necessary information required. Ideally the data is designed to work within a coding environment rather than excel. We aim to write some additional code that will enable uses to extact the data into its different constituent parts (eg. Ages, data) to run new ages depth models and plot data using newly developed models. This code will be included in a subsequent publication utilising the records for continental palaeofire compilation.

We have included code in the Supplement that is designed purely for SahulCHAR and can be used to access the data.

Reviewer 2

The manuscript by Rehn, Cadd et al. presents a new collection of paleofire records gathered across Australia, New Guinea and New Zealand, a globally relevant area of intensifying fire regimes under climate and societal change. To better understand the regions long-term fire history, the role of climate and vegetation change in driving fire as well as the impacts of fire regime changes, it is essential to go beyond directly observed fires (with written or remotely sensed observation being available since few decades to centuries) and include sedimentary proxy datasets such as charcoal and black carbon over the last millennia in the discussion about dealing with fire in this area or for data-model comparisons. To better disentangle local from more large-scale drivers and impacts and for comparing with modelled fire-vegetation interactions, syntheses of multi-site paleofire records have already proven to become highly relevant to improve our understanding and the simulation of fire regime changes (see references in the manuscript, Van Marle et al., 2017, GMD). The manuscript is well-written and understandable and highly suitable to be published in ESSD after some clarifications.

Thank you for your review. We are pleased to see that other researchers can see the benefit of this work.

The SahulCHAR collection now provides a highly relevant harmonized compilation of newly digitized and author-provided datasets, and includes also some updated records already stored in globally available paleorecord compilations. Of the latter, most data derived from the Global Paleofire Database (GPD, formerly Global Charcoal Database, GCD), which is specifically dedicated to provide long-term access to paleofire data but lacks data curation. The authors acknowledged that the records stored there are of different quality, though more detailed information on what exactly has been changed for the GPD records that became part of the SahulCHAR collection is missing (e.g. in terms of x% of the records required coordinate updates, y% required correction of site descriptions or contained missing values in comparison to the original publication etc). This is a matter of ongoing debate.

Our response to this comment was largely addressed in the comments to Reviewer 1. We improved the clarity of the text surrounding the discussion of previous databases and how they were incorporated into the new SahulCHAR database. We have included the additional information requested by both reviewers regarding the transfer of datasets from GPD and how each individual site was incorporated into SahulCHAR. Very few sites were transferred verbatim and each record was critically appraised, with additional information or corrections added when necessary.

The manuscript also contributes to the debate on how to harmonize the high diversity of charcoal and black carbon measurement units in publications and existing databases (see Dietze et al., 2024: First steps towards integrating the Global Paleofire Database with Neotoma | PAGES Magazine 2024) – a crucial step before GPD datasets can be integrated into the Neotoma Paleoecological Database. Especially the separation of the measurement unit (e.g. counts or a concentration in cm³) from the charcoal size and to group measurements into classes of minimum and maximum size classes is highly attractive. Here it would be great to also learn how many of the individual measurements overall in the data collection would fall in which class, e.g. as an extension of Fig. 4. In addition, figure 4 that provides an overview of the characteristics of the data compilation, could also be extended to learn which time periods are covered by the compiled records – that would surely increase its attractivity for users.

This was a difficult issue we grappled with when incorporating the different reported metrics of charcoal counts. Whilst we can see there might be some users of the database that would like to know the number of individual measurements for each charcoal measure, this is a large amount of data that would be largely meaningless without knowing which individual records this data comes from. We believe the number of records that use the different charcoal measures is sufficient for this manuscript, primarily designed to provide an overview of the SahulCHAR database.

Including the age range of the different records is a good idea. We have modified the map figure to show the variety of ages of the different records in the database. We believe this is the best way to display this data without adding additional figures or panels.

The SahulCHAR collection is hosted by OCTUPUS, a database platform that is briefly described. To improve the understanding of the "why this platform" and of the overall structure illustrated in Fig. 1 that shadows out other Sahul-compilations, it would help the reader if the authors would briefly describe the purpose and framing of this platform. Also briefly, and more for non-database experts among the users, how data sets would SahulCHAR records be "fully integrated" to other Sahul partner collections (e.g. via sites or even samples)? Please, briefly describe the links in Fig. 1.

We have included some additional description of the OCTOPUS database and how the different collections will interact. We also refer the reviewer and other readers to the other papers published that describe the OCTOPUS database.

Following on that: the authors mention that it has some compatibility with the GPD by using similar categories of metadata and dataset categories, but the structure of OCTOPUS is different, also compared to Neotoma, to which it might contribute in the future, as a bit vaguely described in the "conclusion". As this is a project-based compilation that encourages also for updates and future data submissions: How do the authors envision a (technical) connection between the versions of SahulCHAR and the GPD as a new Constituent Database in Neotoma, to avoid data loss and parallel datasets that may promote insecurity about data quality and originality as we are currently facing with the existing multiple open paleofire data compilations?

We have not determined how exactly the SahulCHAR database will become a constituent database in Neotoma, however several of the authors are working with the global Palaeofire community and Neotoma managers to determine how this will occur morning forward. The selection of size classes, data metrics and metadata kept the future integration with Neotoma in mind, with the hope to reduce any additional future workload in incorporating the data into Neotoma.

Some minor suggestions:

- Consider to define at the beginning what is meant by "black carbon" (multiple methods are behind that may give different measurement units, which not only include classical "count" data as common for charcoal) and briefly state if or how black carbon records were added to a size-related field such as CHARSIZE_U

  We have added a definition of charcoal and black carbon to the introduction and further refined the text that specifies the method of charcoal quantification.

  "The structure of SahulCHAR differs from the GPD in its approach to char sizes and measurement units. SahulCHAR specifies the method of charcoal quantification (CHARMETHOD) for each record, with 11 methodologies for measuring charcoal and black carbon particles included.":

  I am aware that BASIN refers to the depocenter of the sedimentary archive, correct? If I understand right, it is the size of the polygon of the METASITE, yet I was a bit confused by the example in Tab 1 where the CATCHMENT is smaller than the BASIN. Also consider to add a note in the text on which basis the polygons were drawn (esp. year or months) as especially lake extents could vary seasonally and interannually.

  Thank you for noticing this issue with the example in Table 1. The site used in the text was actually incorrectly reported in the original paper. This has been rectified in the database and now in the manuscript as well.

  METASITE is a bounding box geometry around a semantic entity, in the case of SahulChar, a geographic feature such as a lake. Both BASIN and CATCHMENT classes and size values were taken from GDP. We did not calculate those values for the non-GPD sites; therefore we have n=294 'ND' values for CATCHSIZE. Re small catchment, but large basin – indeed, those classes are binned differently (Parent table fields & values — OCTOPUS database documentation 1.0 documentation), meaning that a small CATCHMENT can fit into a large BASIN. These two classifications are not logically related, therefore there may be some odd variations.

- For all field in Tab 1-3 that refer to a list, consider to add a hyperlink to the respective list, as done for BIOME, for consistency and easier understanding.

  We have updated the captions for Tables 1-3 and included a hyperlink to the pages with the table definitions.

---

## Author Response (AR2)

Suggestions for revision or reasons for rejection

Thanks for carefully considering the reviews and revising this manuscript. The revisions have improved the quality of the text, I like the additional information in Fig. 2.

Now only some few details might benefit from re-phrasing or clarification to improve understanding of the dataset and the text for future users.

l. 73: For reference to the current GPD version, I would suggestion to only refer to https://www.paleofire.org plus adding the date (or period) when the data was retrieved/downloaded, as Power et al., 2010 refer to version 1 of the Global Charcoal Database.

We have amended this citation and added the reference and retrieval date to the reference list.

I am not fully convinced concerning your explanation in the response to one review: "All Neotoma and PANGAEA data were entered with the source as 'author' because those databases explicitly show that those records were submitted directly by the authors."

A lot of datasets in the GPD have also been added by authors, which might, however, not be clear in any case. I am also bit irritated by your addition in the text "Original char data from any source are classified in SahulCHAR as CHARSOURCE = 'author'. Data sourced from another database either from digitization data or unknown origins are classified as 'paleofire database', and data digitized from published diagrams for SahulCHAR are classified as 'digitized'" Here, especially the "any" in the first sentence is unclear (in respect to the preceding sentence) and "digitization data" vs. "data digitized".

I would suggest to classify: 1) all datasets that derive from a paleofire database as such, 2) only those that were so far not at all included in any paleofire database and provided by a data author as "author" and all that you newly digitized as "digitized".

We have amended this in the database. For sites with material directly obtained from author for this study it is now referred to as CHARSOURCE = 'author'. Sites where data was obtained from a database but were originally author submitted are now CHARSOURCE = 'paleofire database (author)', with the DATASRCDSC listed as "Author submitted data from Global Paleofire Database or PANGEA Database". Sites obtained from GPD but not known if author submitted are still listed as CHARSOURCE = 'paleofire database'.

Concerning your addition of the following: "we have not included new chronologies" (l. 185), please specify in the text or in Table 3, where the "estimated age for sample" entry derives from then. From the original records of the "CHARsource" you specified before? Maybe just a short reference in the text to Table 3 could help here.

We have added to the text in line 183 "Original chronologies produced by original authors are included, yet it is recommended that re-calibration of age-depth models is conducted using the most appropriate and up to date methods for records included in SahulCHAR at time of use." And added to table 3 that the Estimated age for sample is from original publications.

In the supplement, please add a clear comprehensive caption for the provided table S1 and refer to this table in the text where you mention that records from the GPD were included after careful checking and corrections (e.g., end of chapter 1 or chap 2.2.1 Charcoal and black carbon metadata, that should actually be chapter 2.2.2 as a 2.2.1 is already existing). Please add a caption for the R script in the supplement, too, and also refer to it in the main text to make users aware of it.

We have included references to the Supplementary information and this additional text in the Data Summary section:

"SahulCHAR is hosted on the OCTOPUS platform (https://octopusdata.org) and can be accessed directly from the web interface (https://octopus-db.github.io/documentation/usage.html#web-interface) or accessed via Web Feature Service (https://octopus-db.github.io/documentation/usage.html#web-feature-service). The WFS data can be accessed directly through GIS or R software (see supplement for example code)."